# Polyphenols Rich Diets and Risk of Type 2 Diabetes

**DOI:** 10.3390/nu13051445

**Published:** 2021-04-24

**Authors:** Andrea Da Porto, Alessandro Cavarape, GianLuca Colussi, Viviana Casarsa, Cristiana Catena, Leonardo A. Sechi

**Affiliations:** Clinica Medica, University of Udine, 33100 Udine, Italy; alessandro.cavarape@uniud.it (A.C.); gianluca.colussi@uniud.it (G.C.); viviana.casarsa@gmail.com (V.C.); cristiana.catena@uniud.it (C.C.); leonardo.sechi@uniud.it (L.A.S.)

**Keywords:** polyphenols, diet, dietary patterns, diabetes

## Abstract

Type 2 diabetes is an increasing health concern worldwide. Both genetic and environmental risk factors as improper dietary habits or physical inactivity are known to be crucial in the pathogenesis of type 2 diabetes. Polyphenols are a group of plant-derived compounds with anti-inflammatory and antioxidant properties that are associated with a low prevalence of metabolic conditions characterized by insulin resistance, including obesity, diabetes, and hypertension. Moreover, there is now full awareness that foods that are rich in phytochemicals and polyphenols could play an important role in preserving human cardiovascular health and substantial clinical evidence indicates that regular dietary consumption of such foods affects favorably carbohydrate metabolism. This review briefly summarizes the evidence relating dietary patterns rich in polyphenols with glucose metabolism and highlights the potential benefits of these compounds in the prevention of type 2 diabetes.

## 1. Introduction

The chemical structure of phenols is characterized by an aromatic ring with at least one hydroxyl group and can vary [1] from simple molecules to complex polymers with high molecular weight. Although there is some disagreement concerning their categorization, a classification that has been commonly used in the medical literature subdivides phenols in two main groups: Flavonoids and non-flavonoid polyphenols [2,3]. The flavonoid group comprises flavanones, flavones, dihydroflavonols, flavonols, flavan-3-ols, isoflavones, anthocyanidins, proanthocyanidins, and chalcones, all compounds with a C6-C3-C6 structure. Non-flavonoid polyphenols are classified on the basis of their carbon skeleton into different subgroups: Simple phenols, benzoic acids, hydrolysable tannins, acetophenones, phenylacetic acids, cinnamic acids, lignans, coumarins, benzophenones, xanthones, stilbenes, and secoiridoids [4].

Hundreds of different polyphenols are found in plant-based foods (Appendix A
Table A1) including fresh vegetables (e.g., broccoli, onion, cabbage, garlic, asparagus, carrots), fruits (grapes, pears, apples, cherries, and various berries), legumes (soybean), and cereals (whole grains). Other relevant dietary sources of polyphenols are identified in plant-derived beverages such as chocolate, coffee, tea, red wine, or extra-virgin olive oil [5,6]. There is now full awareness that foods that are rich in phytochemicals and polyphenols could play an important role in preserving human cardiovascular (CV) health [7], and substantial clinical evidence indicates that regular dietary consumption of such foods favorably affects carbohydrate metabolism [8]. This review briefly summarizes the evidence relating to dietary patterns rich in polyphenols with glucose metabolism and highlights the potential benefits of these compounds in the prevention of type 2 diabetes. 

## 2. Dietary Pholyphenols and Diabetes Risk

High amounts of polyphenols are present in some diets such as the Mediterranean and the plant-based diet. The Mediterranean diet essentially refers to a dietary pattern rich in whole grains, fruits and vegetables, legumes, nuts, fish, olive oil, and with moderate wine consumption. It has been estimated that adherence to a Spanish Mediterranean diet led to a daily intake of polyphenols between 2590 and 3016 mg/die [9]. Plant-based diets involve eating plenty of vegetables, fruits, and cereals and a low amount of animal products; among plant-based diets, Pesco-Vegetarian, Semi-Vegetarian, and Lacto-ovo-Vegetarian diets contain the greatest amounts of polyphenols [10]. It is known that both the Mediterranean and plant-based diets are deemed “healthy diets” because, due to their content in polyphenols, they have been shown to improve carbohydrate metabolism and glucose tolerance [11]. Therefore, the potential benefit of these diets for prevention of type 2 diabetes has been tested in intervention studies. A prospective study that included 6798 participants from the Rotterdam Study demonstrated that high adherence to a plant-based versus animal-based diet was associated with a lower risk of insulin resistance, prediabetes, and diabetes, even after adjusting for confounders such as sociodemographic and lifestyle factors (food supplements, medications use, physical exercise) [12]. Similar findings were obtained from the pooled analysis of three big databases (Nurses’ Health Study, Nurses’ Health Study 2 and Health Professionals Follow-Up Study) [13] in which a healthful plant-based diet rich in whole grains, fruits, vegetables, nuts, legumes, vegetable oils, and tea/coffee was associated with substantially lower risk of developing type 2 diabetes during the follow-up. Furthermore, as demonstrated in a meta-analysis that included data from 307,099 patients [14], plant-based diets that are enriched with healthful plant-based foods are beneficial for the primary prevention of type 2 diabetes. On the other hand, at least five large prospective studies reported a substantially lower risk to develop type 2 diabetes in healthy or at-risk subjects who ate a Mediterranean diet [15]. To date, the relative contribution of single components of the Mediterranean or the plant-based diets to type 2 diabetes prevention is not clearly understood.

In this context, the role of specific dietary components including a group of polyphenols has been extensively studied in the last decade. A randomized controlled trial reported that a diet naturally rich in polyphenols improves glucose metabolism in individuals at high risk of diabetes [16]. A meta-analysis of six prospective cohorts that involved 284,806 participants suggested that an increase in the total dietary flavonoids intake by approximately 500 mg/day is associated with a significant decrease of the risk to develop type 2 diabetes [17] and, consistently, another meta-analysis of 18 prospective studies [18] showed that diets rich in polyphenols, particularly flavonoids, play a role in the prevention of type 2 diabetes. A large epidemiological study showed that anthocyanins-rich diets with substantial consumption of specific whole fruits, including blueberries, grapes, and apples, was significantly associated with a reduced risk of diabetes [19]. In a 3-month randomized, double-blind, placebo-controlled trial of subjects with prediabetes or new onset diabetes, purified anthocyanins moderately reduced glycated hemoglobin [20]. Another recent study has indicated the potential benefit of anthocyanin-rich mixed berry preparations on post-prandial glucose and insulin response [21] thereby suggesting an improved insulin sensitivity. Further evidence comes from a recent metanalysis of 37 randomized controlled trials that demonstrates that consumption of anthocyanins for more than 8 weeks in doses of more than 300 mg/day significantly decreases fasting and post-prandial glucose, glycated hemoglobin, and HOMA-index in overweight/obese individuals [22]. In addition, in a Korean study involving 4186 individuals, the consumption of flavonols and flavones was directly associated with insulin sensitivity among male subjects [23]. According to all these studies, it could be stated that current evidence indicates the benefits of polyphenols-enriched diets, particularly with flavonoids, on insulin sensitivity that could translate to a significantly reduced risk to develop type 2 diabetes. It is appropriate, however, to mention that other studies have not confirmed this evidence [24] and a prospective, cross-sectional analysis of 38,018 women reported did not observe any relationship between flavonols or flavones dietary intake and the risk of type 2 diabetes, and the intake of anthocyanidins was not linked to the risk of diabetes [25]. Inconsistencies among the results of these studies might be explained by substantial differences in the populations included the use of different types of polyphenols sources, adherence to dietary prescriptions with possible assessment of dietary intake, and the duration of exposure.

## 3. Specific Dietary Sources of Polyphenols and Diabetes Risk

### 3.1. Grains and Soy

Regular consumption of whole-grain products has been associated with a significant reduction in the risk of type 2 diabetes and lower CV mortality in many prospective studies [26,27,28]. Inverse associations between whole-grain intake, fasting insulin, and insulin sensitivity have been reported [29], although differences may occur due to grain variety. For instance, whole-wheat breads are associated with lower glycemic responses than white bread [30], and this has been attributed, at least in part, to the polyphenols content in the bran fraction of whole wheat [31]. It was demonstrated that ferulic acid contained in whole grains decreases cellular glucose uptake into epithelial cells in the small intestine and therefore might account for the attenuation of postprandial hyperglycemia by inhibition of glucose absorption [32]. Although dietary intake of soy protein and soy isoflavones was found to be inversely related to new onset type 2 diabetes in two meta-analyses of prospective studies [33,34], the evidence currently available on the possible benefits of soy on glucose metabolism is still preliminary and needs further investigation. 

### 3.2. Fruits and Vegetables

Apples are rich in bioactive polyphenols and were found to decrease postprandial blood glucose in two small randomized controlled trials [35,36] conducted in healthy subjects. However, a recent study of subjects who consumed apples on a regular basis did not show any improvements of fasting blood glucose or serum insulin when compared to controls [37].

It is well known that berries are rich in polyphenols and contain a wide variety of phenolic molecules, ranging from phenolic acids (hydroxybenzoic and hydroxycinnamic acids), flavonoids (anthocyanins, flavonols, and flavan-3-ols) to polymerized compounds (proanthocyanins and ellagitannins) [38]. In a study conducted in prediabetic subjects with an oral glucose test, raspberry reduced the post-glucose load peak serum glucose and insulin levels and the area under the curve of glucose and insulin response [39]. In an elegant double-blind randomized controlled trial, consumption of 333 mg/day of polyphenols from strawberries and cranberries for 6 weeks improved insulin sensitivity and decreased compensatory insulin hypersecretion in patients with overweight [40]. Conversely, findings of other studies indicate that daily consumption of cranberry beverages for 8 weeks does not improve insulin sensitivity [41] and consumption of 140 g/day of blueberries with a carbohydrate breakfast meal does not affect glucose metabolism, nor the gastrointestinal hormones Glucagon Like Peptide 1 (GLP-1), Gastro Inhibitory Peptide, and peptide YY.

Also, pomegranate has an elevated content of bioactive polyphenols, and possible glucose-lowering effects of this fruit have been reported in animal studies [42], but intervention studies in humans have generated controversial results. In a randomized, cross-over, controlled study on healthy volunteers, beverages containing pomegranate polyphenols (mainly tannins) were found to reduce the postprandial glycemic response to a bread meal [43]. However, a meta-analysis of 16 randomized controlled trials showed that pomegranate intake did not have any favorable effect on glucose metabolism and insulin sensitivity [44].

Among additional promising antidiabetic compounds, citrus flavonoids were reported to modulate the intracellular pathways related to cell glucose uptake and insulin sensitivity and therefore they might have relevance in the prevention of diabetes, although this role needs to be verified in appropriately designed human studies [45].

Because of its abundant content of polyphenols, kiwifruit was sought to have beneficial metabolic effects, mostly relevant for carbohydrate metabolism. However, a meta-analysis of five randomized controlled trials has reported no favorable metabolic effect of kiwifruit in patients with cardiovascular risk factors [46].

Onion bulbs are one of the greater sources of dietary flavonoids and contribute significantly to the overall intake of flavonoids in common diets [47]. Preclinical studies suggested that onion bulbs and its active components may be considered as prophylactic or therapeutic agents against diabetes through a variety of mechanisms including an antioxidant, α-glucosidase, and α-amylase inhibitory effect, up-regulation of adiponectin receptors, reduction of insulin resistance and glucose intestinal absorption, elevation of hepatic and muscular glycogen content, increase of insulin secretion, and phosphorylation of AMP-activated protein kinase [48]. The metabolic effects of onion consumption were tested in a randomized controlled trial conducted in overweight/obese women with polycystic ovary syndrome reporting no significant difference in fasting blood sugar after 8 weeks [49]. In another randomized, placebo-controlled, 8-week trial, dietary raw yellow onion at the doses of 100 to 160 g/day improved doxorubicin-related insulin resistance and hyperglycemia in women with breast cancer [50].

Broccoli and broccoli sprouts are important sources of bioactive compounds including isothiocyanates, glycosylates, flavonoids, and phenols, all compounds that may improve insulin resistance. In a small randomized clinical trial, the consumption of 10 g/day of broccoli sprouts for 4 weeks resulted in a significant decrease in fasting serum insulin concentration and HOMA-index [51]. In another open trial, a combined meal of cooked broccoli and mashed potato had significantly lower serum glucose and insulin responses in comparison to potato eaten alone, suggesting that broccoli favorably affect glucose homeostasis [52].

In summary, polyphenols are contained in many fruits and vegetables whose effects could contribute significantly to an improvement of glucose metabolism and insulin sensitivity, possibly leading to the prevention of type 2 diabetes. However, the evidence obtained in intervention studies is still controversial for many types of fruits and vegetables and will need more extensive investigation before these dietary recommendations could be considered as mandatory in subjects at risk of developing type 2 diabetes. 

### 3.3. Olive Oil 

Extra-virgin olive oil is one of the components that most differentiates the Mediterranean diet from other dietary patterns. Phenolic compounds such as oleuropein and hydroxytyrosol, flavonoids, especially flavones, and lignans are abundant in extra-virgin olive oil [53]. The phenolic composition of olive oil ranges from 50 to 800 mg/L depending upon several factors including, among others, variety, cultivation techniques, degree of ripeness, and climate [54].

The effects of olive oil consumption on prevention of type 2 diabetes have been tested in many randomized clinical trials. The PREDIMED trial examined the effects on glucose metabolism of a Mediterranean diet with extra supplements of extra-virgin olive oil [55] and showed that this diet lowered the risk of type 2 diabetes by 40% in patients with high cardiovascular risk in comparison to a control group. Other analyses of the PREDIMED trial showed a significant reduction of new-onset type 2 diabetes in elderly individuals with the highest dietary intake of phenols and an inverse correlation between the degree of obesity and phenol consumption [56].

Recently, the PREDIABOLE study has tested the effect of increased intake of oleanolic acid on new-onset diabetes in prediabetic subjects [57]. In another randomized cross-over study of 30 patients with impaired fasting glucose, a meal containing 10 g of extra-virgin olive oil decreased blood glucose levels probably by inhibition of Dipeptidyl-peptidase-4 activity and increase of insulin and GLP-1 secretion [58]. The potential benefits of olive oil for the prevention of type 2 diabetes have been confirmed in a meta-analysis of 29 prospective studies in which the highest olive oil dietary intake category showed a 16% reduction in the risk of type 2 diabetes [59]. Thus, current evidence suggests that diets enriched with olive oil might prevent new-onset diabetes, an effect that, at least in part, could be attributed to the polyphenol content of oil. 

### 3.4. Red Wine 

A moderate and regular wine consumption is another typical component of the Mediterranean diet. Polyphenols are highly concentrated in red wine and come from the skin and seeds of grapes extracted during fermentation. The amount of polyphenols in a glass of red wine is approximately 200 mg compared with the 30 mg contained in a glass of white wine. Results of a small randomized controlled trial involving men at high cardiovascular risk support the hypothesis of a beneficial effect of the non-alcoholic fraction of red wine (mainly polyphenols) on insulin-resistanc; moreover, polyphenols of red wine were reported to be beneficial on and plasma lipids concentrations [60].

Resveratrol is a stilbenoid present in a variety of plants, such as Polygonum cuspidatum, grapes, and peanuts, and is considered one of main polyphenolic compounds of red wines. Experimental animal and human studies have shown that resveratrol has many beneficial metabolic effects due to its ability to improve insulin sensitivity and beta-cell function [61]. A meta-analysis of 30 studies that evaluated the effect of resveratrol on glucose, insulin, and glycated hemoglobin levels and insulin resistance (HOMA-index) reported a significant reduction of fasting glucose and insulin levels but no significant changes in glycated hemoglobin and HOMA-index. 

In summary, there is currently limited evidence of the effect of red wine polyphenols on glucose metabolism and insulin sensitivity. More studies would be needed to better understand the antidiabetic properties of resveratrol, as well as to define the therapeutic potential of other stilbenes in the prevention of type 2 diabetes.

### 3.5. Tea, Coffee, and Chocolate

Encouraging data regarding the capacity of polyphenols to prevent type 2 diabetes were obtained in studies that investigated the effects of phenolic acids contained in tea and coffee in China and Japan [62,63]. In a randomized, controlled, cross-over study of obese subjects, a daily dose of flavonoid-rich black tea decreased postprandial glucose and insulin concentrations [64]. Green tea was superior to metformin in improving glycemic control in a randomized controlled trial conducted in 120 non-diabetic overweight women [65]. A meta-analysis of 17 randomized studies showed that green tea had favorable effects on glucose metabolism, decreasing fasting glucose and glycated hemoglobin concentrations [66]. Conversely, another meta-analysis of seven trials reported that consumption of green tea did not affect fasting plasma glucose and insulin levels, post-load plasma glucose, glycated hemoglobin, and HOMA-index in subjects at risk of new-onset type 2 diabetes [67].

A recent meta-analysis of 28 prospective studies on 1,109,272 non-diabetic subjects showed a 33% lower incidence of type 2 diabetes in habitual coffee drinkers (one to six cups of coffee/day, with and without caffeine) than in non-consumers [68]. However, in another study conducted in healthy subjects, coffee beverages with different contents of chlorogenic acids did not show any significant effect on glucose or insulin responses [69].

Cocoa flavanols are present in good amounts in dark chocolate, with a content of catechin and epicatechin (estimated to be approximately 20 times higher than in tea [70]) that was reported to have some beneficial effects on glucose metabolism. Furthermore, a meta-analysis of 42 randomized controlled trials with acute or chronic consumption of chocolate or cocoa reported an improvement in insulin sensitivity associated with a decrease in serum insulin [71]. Cocoa-related polyphenol intake may also result beneficial in reducing appetite and body weight although this effect needs further investigation [72].

## 4. Conclusions

In conclusion, although many epidemiological studies suggest that diets based on foods rich in polyphenols are associated with a significant reduction in the risk of developing type 2 diabetes, findings of studies on specific compounds are still controversial (Appendix A
Table A2). This might be due to broad differences in the populations included in studies, the types of food, the assessment of dietary intake, and the duration of exposure. Furthermore, foods are a source of several types of polyphenols so the potential benefits on glucose metabolism are more likely to be mediated by the mixture and integration of them than the activity of a single compound. Thus, to date, it is not possible to draw definitive conclusions on the effects of specific polyphenols on glucose metabolism and on their possible role in prevention of type 2 diabetes. More evidence obtained in accurately designed dietary studies will be needed before polyphenol-rich foods could be includes as a mainstay of the dietary prescription in subjects at risk of type 2 diabetes.

## Data Availability

Not applicable.

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
