# Peer review of "Polyphenols Rich Diets and Risk of Type 2 Diabetes"

_nutrients, 2021, doi:10.3390/nu13051445_

Round 1
Reviewer 1 Report
Da Porto et al. discussed the dietary polyphenols and diabetes risk and specific dietary sources of polyphenols and their effects on type 2 diabetes. However, the author listed the results from literature but without their own analysis. In addition, the preclinical studies that the author cited lack essential experimental details. Also, it is necessary to include a table to list all the preclinical trials mentioned in the manuscript. Further, several review articles on a similar topic have been published elsewhere (Guasch-Ferré, et al. Oxidative medicine and cellular longevity 2017; Del Bo et al. Nutrients, 2019; Cao et al, Critical reviews in food science and nutrition 2019).
Several questions on each section
Dietary Polyphenols and Diabetes Risk
- The author discussed the dietary polyphenols and risk of type 2 diabetes, but several key points of dietary polyphenols were not provided. For example:
L44 Please give more accurate data about the amount of polyphenols from the plant-based diet and Mediterranean diet?
L58 What is the plant-based food? any ingredients or polyphenols percentage?
- contrary results have been described, but the author did not give an explanation/ analysis of the contrary results: for example,
L90-95 flavonols or flavones consumption and diabetes risk
Specific Dietary Sources of Polyphenols and Diabetes Risk
- The author discussed several dietary sources of polyphenols, but it would be more logical if describe the type of polyphenols in the food source first.
- Please check and provide the full form before providing the abbreviated form.
Reviewer 2 Report
The submitted review is focused on the relation between dietary patterns rich in polyphenols with the risk of type 2 diabetes. In my opinion the paper is valuable and important in some areas of diabetes research.
Listed below are some suggested changes:
- the authors should consider the insertion of a table summarizing the commented data
- Roman numeration definitely makes it more difficult to follow the references
- references section: the literature positions should be given in one style- abbreviations should be defined at first mention and used consistently thereafter (e.g. CV, DPP4, GLP-1)
- minor typing errors (e.g. double dots) should be removed
Round 2
Reviewer 1 Report
Thanks for the author's revision. I do not have any further questions/comments regarding the manuscript.